# Preparation and Properties of Different Polyether-Type Defoamers for Concrete

**DOI:** 10.3390/ma15217492

**Published:** 2022-10-25

**Authors:** Min Qiao, Jingzhi Wu, Nanxiao Gao, Guangcheng Shan, Fei Shen, Jian Chen, Bosong Zhu

**Affiliations:** 1State Key Laboratory of High Performance Civil Engineering Materials, Jiangsu Sobute New Materials Co., Ltd., Nanjing 211103, China; 2Bote New Materials Taizhou Co., Ltd., Taizhou 225474, China

**Keywords:** polyether, defoamer, bubble, air void, mortar, concrete

## Abstract

In this study, a series of polyether-type defoamers for concrete which consist of the same alkyl chain (hydrophobic part) but different polyether chains (hydrophilic part) was prepared, and the structure–property relationship of the defoamers was investigated for the first time. Using oleyl alcohol (OA) as the starting agent (alkyl chain), the polyether defoamers with different polyether chains were prepared by changing the amount and sequence of ethylene oxide (EO) and propylene oxide (PO) units. The properties of different defoamers were tested in aqueous solutions, and fresh and hardened mortars; the structure–property relationship of the defoamers was thus studied. The results indicated that the defoaming capacity of the polyether defoamers decreased with an increased EO amount, and the defoamers linked with both EO and PO units (PO before EO) had a stronger defoaming capacity than those linked with EO only. This study is beneficial for the development and applications of novel synthetic polyether-type defoamers for concrete.

## 1. Introduction

As one of the most important engineering materials in the construction industry, concrete has been widely used with the rapid development of the world’s market economy and the increase in construction projects [1,2,3,4,5,6,7,8,9]. However, there are still many problems in the practical engineering applications of concrete, such as resistance to chloride diffusion, slump loss, early-age plastic shrinkage and cracking, expansion of the U-type expansive agent, and hydration heat regulation of mass concrete. These problems are mainly because of air bubbles with large sizes and in excessive amounts, which are harmful and formed in the process of fresh concrete mixing [10,11,12]. The reduction of these harmful bubbles is thus highly desirable to improve the performance of concrete [13,14,15,16]. Defoamer is a chemical admixture which has been widely used to inhibit and eliminate the harmful, large bubbles in fresh concrete [17]. By reducing the harmful bubbles and optimizing the pore structure of concretes, defoamers can improve the mechanical properties and service life of hardened concretes [18,19].

The chemical structure of defoamer is an amphiphilic surfactant which consists of a hydrophilic part and a hydrophobic part [20]. From the chemical composition, commonly used defoamers can be divided into four types, including fatty acids, mineral oils, polyethers and silicones [21,22,23,24,25,26,27]. Among them, polyether-type defoamers show the best water solubility and highest compatibility with polycarboxylate (PCE) superplasticizer, and have been most widely used as concrete admixtures [18]. However, synthetic polyethers have only rarely been reported as the defoamers for concrete. The structure–property relationships of polyether-type defoamers in both aqueous solutions and cement-based materials have not been fully investigated, which is of great importance for the development of novel, high-performance defoamers for concrete.

In our previous study, we reported the polyether-type defoamers (amphiphilic surfactants) consisting of different hydrophobic alkyl chains (linear or branched, C14–25) and the same hydrophilic polyether chain (8 ethylene oxide units), which exhibited excellent defoaming performance [18,19]. The effect of hydrophobic alkyl chains on defoaming properties has been investigated, but the influence of a hydrophilic polyether chain (length and sequence) is still unknown. In this study, a series of polyether-type defoamers consisting of the same alkyl chain but different polyether chains was prepared and investigated for the first time. Using oleyl alcohol (OA) as the starting agent (long alkyl chain), the polyether defoamers with different polyether chains were prepared by changing the amount and sequence of ethylene oxide (EO) and propylene oxide (PO) units. The properties of different defoamers were tested in aqueous solutions, and fresh and hardened cement mortars, and the structure–property relationship was investigated. The results indicated that the defoaming capacity of the defoamers decreased with an increased EO amount, and the defoamers linked with both EO and PO units (PO before EO) had a stronger defoaming capacity than those linked with EO only. This study is beneficial for the applications and development of novel, high-performance polyether-type defoamers for concrete.

## 2. Materials and Methods

### 2.1. Materials

Oleyl alcohol (OA) was obtained from Aladdin Chemical Co., Ltd. (Shanghai, China). Ethylene oxide (EO), propylene oxide (PO) and sodium hydride (NaH) were obtained from Sinopharm Chemical Reagent Co., Ltd. (Shanghai, China). All the reagents are analytical reagents and were used as received. PII52.5 Portland cement was obtained from Jiangnan Xiaoyetian Cement Co., Ltd. (Nanjing, China). The fine aggregate was river sand with a nominal grain size of 0.5–1.5 mm. Polycarboxylate superplasticizer was prepared by Jiangsu Sobute New Materials Co., Ltd. (Nanjing, China).

### 2.2. Methodology

The polyether-type defoamers were prepared as previously reported [28]. The chemical structure of the prepared defoamers was characterized by FT-IR, ^1^H-NMR, and elemental analysis; the results clearly indicate that defoamers of a correct chemical structure and high purity were obtained. The properties of different defoamers were tested in aqueous solutions (surface activity and foaming property), and fresh (air content and bubble size distribution) and hardened (air content, air-void parameter and compressive strength) cement mortars; the structure–property relationship of the defoamers was thus fully investigated.

#### 2.2.1. Synthesis and Characterization of Defoamers

Sodium hydride (1.5 g, 60 wt% dissolved in mineral oil) was dissolved in oleyl alcohol (200 g), and the mixture was poured into an autoclave. The autoclave was vacuumized until the pressure was down to −0.1 MPa, then heated up to 120–130 °C. Different amounts of ethylene oxide (132–328 g), and a mixture of ethylene oxide (132 g) and propylene oxide (174 g), were added for a period of time (0.5–2.5 h), with the reaction temperature of 120–140 °C. The reaction pressure was controlled at 0.3–0.35 MPa. After the reaction pressure decreased to 0.1 MPa, the mixtures were cooled to room temperature; then, the defoamers were obtained as yellow liquids (melting points < 4 °C). 

Infrared spectra (FT-IR) were obtained with a Nicolet Impact 410 FT-IR Spectrometer (Thermo Fisher Scientific, Waltham, MA, USA). The FT-IR spectra were scanned over the wave number range of 500–4000 cm^−1^ using KBr as the carrier. Nuclear magnetic resonance spectra (^1^H-NMR) were obtained with a Bruker AVANCE 400 Fourier transform NMR spectrometer (400 MHz, Bruker, Billerica, MA, USA) using CDCl_3_ as the solvent. Elemental analysis results were carried out on a LECO 932 CHNS elemental analyzer (LECO, St. Joseph, MI, USA). The results were as follows: OE1, calculated for C_26_H_52_O_5_ (%), C 70.22 and H 11.79, and found to be C 70.87 and H 11.31; OE2, calculated for C_30_H_60_O_7_ (%), C 67.63 and H 11.35, and found to be C 67.14 and H 10.95; OE3, calculated for C_34_H_68_O_9_ (%), C 65.77 and H 11.04, and found to be C 66.03 and H 10.82; OE4, calculated for C_38_H_76_O_11_ (%), C 64.37 and H 10.80, and found to be C 64.24 and H 11.29; OEP1, calculated for C_38_H_76_O_9_ (%), C 67.42 and H 11.32, and found to be C 67.73 and H 11.82; and OEP2, calculated for C_38_H_76_O_9_ (%), C 67.42 and H 11.32, found to be, C 67.91 and H 11.95. 

#### 2.2.2. Test of Aqueous Solution Samples

Solution samples (50 mL each) of the defoamers of various concentrations were prepared. A platinum loop was put into a sample and then separated again. The surface tension value was measured when the loop was separated from the liquid surface. Surface tension results were obtained with a Krüss K100 surface tensiometer (Krüss, Hamburg, Germany). Solutions of polycarboxylate superplasticizer (20 wt%) and various defoamers (0.1 wt%) were prepared. Each solution was foamed for 2 min with the gas flow rate of 0.3 L/min. Foam heights, bubble photographs and bubble size distributions were obtained with a Krüss DFA100 dynamic foam analyzer (Krüss, Hamburg, Germany).

#### 2.2.3. Test of Fresh Cement Mortar Samples

Totals of 675 g of cement, 1350 g of sand, 216 g of water, 12 g of the solution (20 wt%) of polycarboxylate superplasticizer and various defoamers (0.1 wt%) were mixed, then stirred using a low speed for 3 min and a high speed for another 1 min. The fresh samples were tested. Air contents were obtained with a SANYO direct reading air content tester (SANYO, Osaka, Japan). Spread diameters were tested according to the Chinese National Standard, GB/T 8077-2012 13. Air contents were tested according to the Chinese National Standard, GB/T 50080-2016 15. Bubble size distribution was obtained with an AVA-3000 pore structure analyzer of fresh concretes (Germann Instruments, Copenhagen, Denmark), according to the previously reported method [29]. 

#### 2.2.4. Test of Hardened Cement Mortar Samples

The fresh samples were incubated for 28 days to prepare the hardened cement mortar specimens (100 mm × 100 mm × 100 mm). The specimens were cut into thin slices with a thickness of 10 ± 2 mm, then ground, buffed, cleaned and coated with a fluorescer, successively. After drying, the hardened samples were tested. Air contents, air-void spacing factors and air-void photographs were obtained with an MIC-840-01 pore structure analyzer of hardened concretes (CHUO SEIKI, Tokyo, Japan). Meanwhile, the fresh samples were incubated for 28 days to prepare the hardened cement mortar specimens (4 cm × 4 cm × 16 cm). Compressive strengths were obtained with an AEC-201 mortar strength testing machine (AEC, Shanghai, China). Compressive strengths were tested according to the Chinese National Standard, GB/T 50081-2002 6. Three same samples of each category were prepared and tested, and the average value was recorded.

## 3. Results and Discussion 

### 3.1. Preparation and Characterization of the Defoamers

Using OA as the starting agent (long alkyl chain), the polyether defoamers of different EO amounts (4, 6, 8 and 10, respectively) were prepared, and polyether defoamers of different EO and PO sequences (OA-4EO-4PO and OA-4PO-4EO) were also prepared (Figure 1 and Table 1). The chemical structure of the obtained defoamers was characterized by FT-IR, ^1^H-NMR and elemental analysis. Figure 2 shows the FT-IR spectrum of OA, including a weak absorption peak of -O-H at 3318 cm^−1^, strong stretching vibration absorption peaks of -C-H at 3000–2800 cm^−1^ and the characteristic adsorption peak of -C=C- at about 1460 cm^−1^. By contrast, the spectra of the defoamers show not only the characteristic adsorption peaks of OA, but also another strong adsorption peak of -C-O- at 1104 cm^−1^, suggesting the EO and PO units were linked to OA successfully.

Figure 3 shows the ^1^H-NMR spectra of the defoamers. For all the defoamers, the peaks at δ = 0.8 are the proton peaks of -CH_3_ of OA, and the peaks at δ = 1.2 are the proton peaks of -CH_2_CH_2_- of OA. For OE1–4, the peaks at δ = 3.5–4.0 are the proton peaks of -CH_2_-O- of EO, and the integral areas of the peaks of the polyether chains increased with the increase in the amounts of EO. For OEP1 and OEP2, the peaks at δ = 1.05 are the proton peaks of -CH_3_ of PO, and the peaks at δ = 3.5–4.0 are the proton peaks of -CH_2_-O- of EO and PO. The results clearly indicate that the EO and PO units were linked to OA successfully, and defoamers of a correct chemical structure were obtained.

### 3.2. Surface Activity of the Defoamers

The surface activity of the polyether defoamers was tested firstly. Solutions containing different defoamers of various concentrations were prepared, and the surface tensions of all the samples were tested. Figure 4 shows that the surface tensions of all the samples decreased gradually with an increasing logarithm of the defoamer concentrations. When the concentration of each sample reached its critical micelle concentration (CMC), the surface tension of each defoamer also reached its minimum value (γ_CMC_). The further increase in the concentration of each defoamer induced only a little decrease in the surface tension. Compared with pure water, with a surface tension of 72 mN/m, the surface tensions of the solutions of all the defoamers reached lower values at lower concentrations. The surface tensions of the samples increased with an increased amount of EO units (decreasing hydrophobicity) [30,31,32]. Table 2 shows the CMC and γ_CMC_ values of the defoamers, which were calculated by plotting surface tension against logarithmic concentration, as in the previous report [33,34,35,36,37]. It was observed that both the CMC and γ_CMC_ values increased with the increase in the amount of EO unit, and the defoamer linked with PO before EO (OEP2) gave the lowest CMC and γ_CMC_ values.

### 3.3. Properties of the Defoamers in Aqueous Solutions

The defoaming performance of the defoamers in aqueous solutions was investigated next. Solutions of polycarboxylate superplasticizer mixed with various defoamers were prepared. Each solution was foamed for 2 min with a gas flow rate of 0.3 L/min, and the real-time changes in the foam height were recorded. Since the solution of polycarboxylate superplasticizer contains a large number of unreacted polyether comonomers, which are hydrophilic and can produce excessive air bubbles in solution, adding the defoamers to the solution can eliminate these bubbles effectively. Figure 5 shows that the foam heights of all the samples increased gradually with the foaming time, and then decreased with the incubation time. The maximum foam heights decreased with the decrease in the amount of EO units, indicating that the defoamer containing less EO (higher hydrophobicity) had a higher defoaming ability in aqueous solutions [18,19]. It was also observed that compared with the defoamers linked with EO only, the defoamers linked with both EO and PO caused the foam heights to decay more quickly during the incubation time, and the defoamer linked with PO before EO (OPE2) caused the foam height to decay the most quickly. The results indicated that the defoamer linked with both EO and PO had a stronger defoaming capacity, since PO is more hydrophobic than EO, which exhibits weak hydrophilicity. Compared with OEP1 (OA-4EO-4PO, hydrophobic–hydrophilic–weakly hydrophilic), the structure of OEP2 (OA-4PO-4EO, hydrophobic–weakly hydrophilic–hydrophilic) is closer to that of an amphiphilic surfactant; thus, OEP2 exhibits a stronger defoaming capacity.

Table 3 shows the maximum foam heights and complete defoaming times of all the solution samples. The results indicate that both the maximum foam heights and complete defoaming times decreased with the decrease in the amount of EO units. The defoamers linked with both EO and PO gave the maximum values of the foam heights and lower complete defoaming times, compared with the defoamers linked with EO only, and the defoamer linked with PO before EO (OEP2) gave the lowest values. The results clearly indicated that the defoamer containing less EO had a higher defoaming capacity, and the defoamer linked with PO before EO had the strongest defoaming capacity. Figure 6 shows the photographs of the bubbles of the samples mixed with various defoamers at the total time of 700 s. It was observed that the defoamer containing less EO induced less and a larger size of bubbles at the time of 700 s, and the defoamer linked with PO before EO (OEP2) induced nearly no bubbles. The results further support the conclusions more directly, which suggest that during the incubation time, the defoamers caused more tiny bubbles to burst then merged into large bubbles. 

### 3.4. Properties of the Defoamers in Fresh Cement Mortars

The defoamers in this study exhibited both a high surface activity and defoaming capacity in aqueous solutions. We envisioned that they may also have had a good defoaming capacity in cement-based materials. The defoaming properties of the defoamers in fresh cement mortars were thus investigated. Table 4 shows that compared with the blank sample, the air content of all the samples mixed with the defoamers markedly decreased, and the air content decreased with the decrease in the amount of EO. It can be observed that the defoamers linked with PO before EO (OEP2) gave lower air content (higher bulk density) compared with those linked with EO only. The defoaming properties of the defoamers in fresh cement mortars are highly consistent with those obtained in aqueous solutions. The results tested in fresh cement mortars are also consistent with the results of surface tension, and the defoamer having a lower surface tension value exhibits a higher defoaming capacity. The bubble size distribution of the fresh mortars mixed with various defoamers was also tested. Figure 7 shows that all the defoamers could decrease the air content of the fresh mortars at all the bubble size ranges, and the air contents of the air bubbles at the large sizes (>1 mm) caused a much larger decrease. Since large air bubbles may weaken the mechanical performance of hardened concretes much more strongly [14], the defoamers in this study preferred to inhibit and eliminate the harmful large bubbles in the fresh concretes, which may be advantageous to enhance the mechanical performance of hardened concretes.

### 3.5. Properties of Defoamers in Hardened Cement Mortars

The defoaming properties of the defoamers in hardened cement mortars were also investigated. Hardened specimens mixed with various defoamers were prepared and tested. Table 5 shows that compared with the blank sample, the air content of all the samples mixed with the defoamers obviously decreased, and the air content decreased gradually with a decreasing amount of EO. The sample containing the defoamer linked with PO before EO (OEP2) showed a lower air content value compared with those linked with EO only, which is consistent with the conclusions obtained in both the aqueous solutions and fresh cement mortars. 

An important air-void parameter of hardened concretes, the air-void spacing factor, was discussed [30,31,32]. Table 5 also shows that the samples with lower air content features had a larger air-void spacing factor. Figure 8 shows the air-void photographs of the hardened mortars mixed with various defoamers to observe the air voids more directly. It can be clearly observed that the samples mixed with the defoamer with a higher defoaming capacity produced air voids in lesser amounts and with longer air-void distances. The results further support the conclusions of the air content and air-void spacing factors, which clearly suggest that the defoamers in this study eliminated the excessive harmful air voids and decreased the air content of hardened concretes effectively. 

Unlike the small bubbles (<200 μm) produced by air-entraining agents (good for workability and freeze–thaw durability), the use of polycarboxylate superplasticizer has a tendency to produce an excessive amount of large bubbles in fresh concrete. Both the amount and quality of these entrapped bubbles are not controlled, which may decrease the mechanical performance of hardened concrete [13,14]. Inhibiting and eliminating the excessive harmful bubbles in fresh concretes has been reported to be advantageous in enhancing the mechanical performance of hardened concretes [38,39,40,41,42]. Table 5 and Figure 9 also show the compressive strengths of the hardened mortar samples mixed with various defoamers after incubation for 28 days. It can be clearly observed that compared with the blank sample, the compressive strengths of all the samples mixed with the defoamers were enhanced, and the defoamers with a higher defoaming capacity caused a larger enhancement of the compressive strength. The results suggest that using polyether-type defoamers in this study was highly beneficial in enhancing the compressive strength of hardened concretes.

## 4. Conclusions

In summary, a series of polyether-type defoamers consisting of the same alkyl chain but different polyether chains was prepared and investigated for the first time. Using OA as the starting agent (hydrophobic alkyl chain), polyether defoamers with different hydrophilic polyether chains were prepared by changing the amount and sequence of EO and PO units. The properties of different defoamers were tested in aqueous solutions, and fresh and hardened cement mortars, and the structure–property relationship was investigated. The results indicated that the defoaming capacity of the polyether defoamers decreased with the increase in the EO amount, and the defoamers linked with both EO and PO units had a stronger defoaming capacity than those linked with EO only. The structure of the defoamer linked with PO before EO (hydrophobic–weakly hydrophilic–hydrophilic) is closer to that of an amphiphilic surfactant, which exhibits a stronger defoaming capacity. This study is highly beneficial for the application and development of novel synthetic polyether-type defoamers for concrete.

## Figures and Tables

**Figure 1 materials-15-07492-f001:**
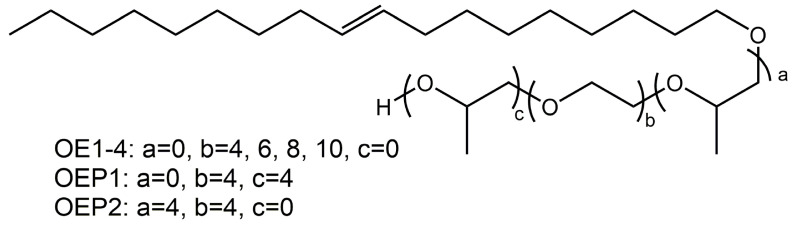
Chemical structure of the prepared defoamers.

**Figure 2 materials-15-07492-f002:**
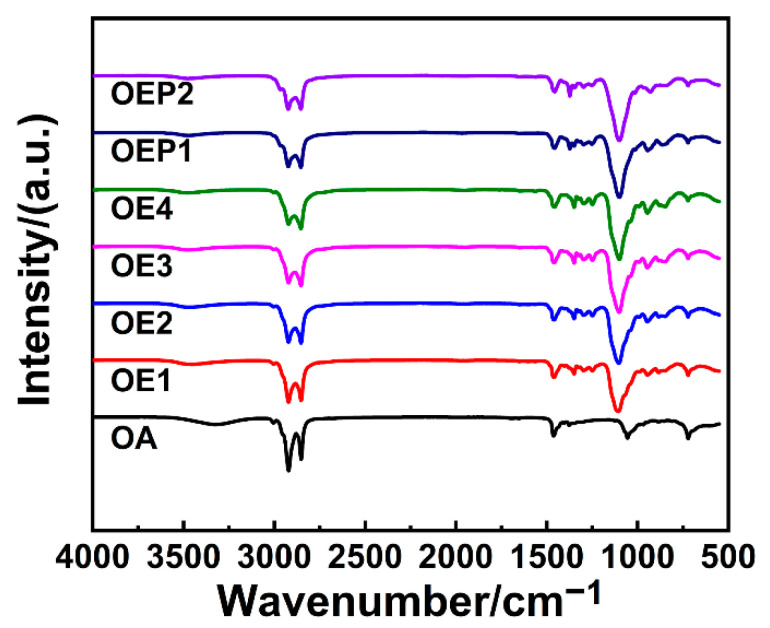
FT-IR spectra of the prepared defoamers.

**Figure 3 materials-15-07492-f003:**
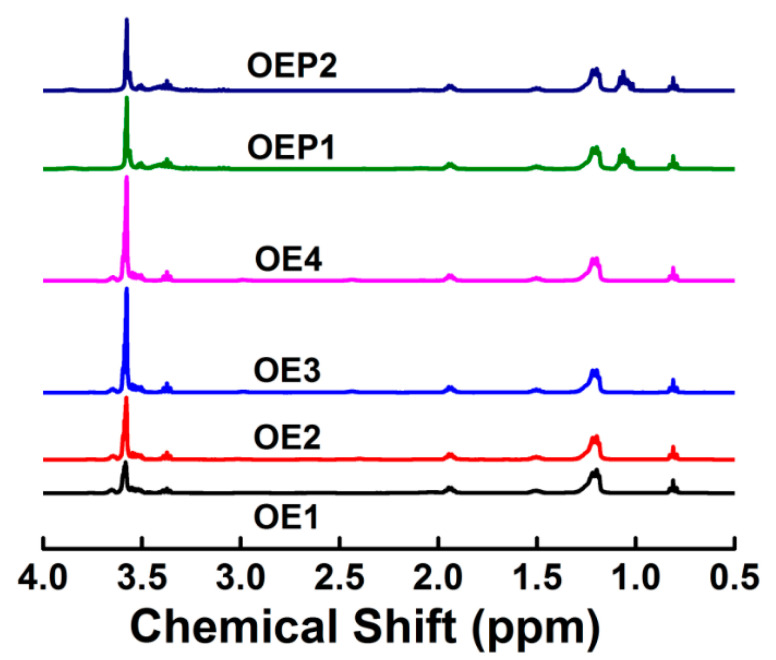
^1^H-NMR spectra of the prepared defoamers.

**Figure 4 materials-15-07492-f004:**
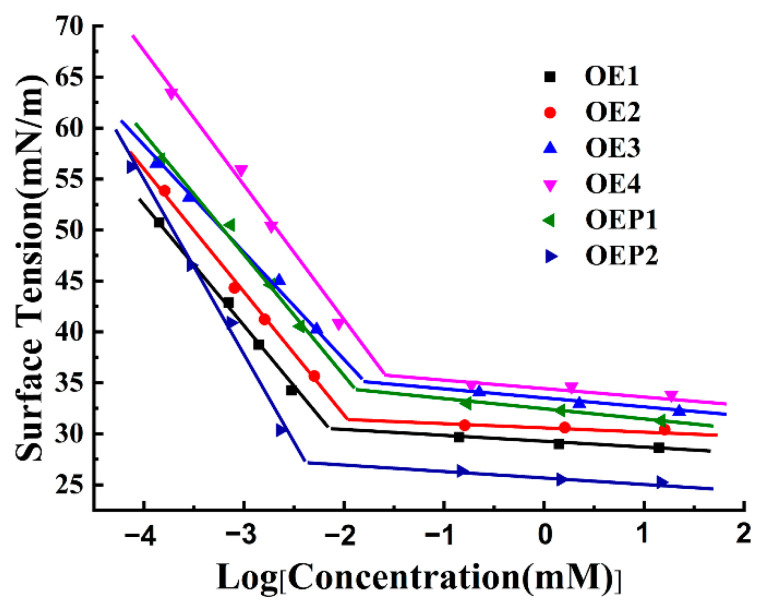
Plot of the changes in surface tension vs. the logarithm of the defoamer concentration.

**Figure 5 materials-15-07492-f005:**
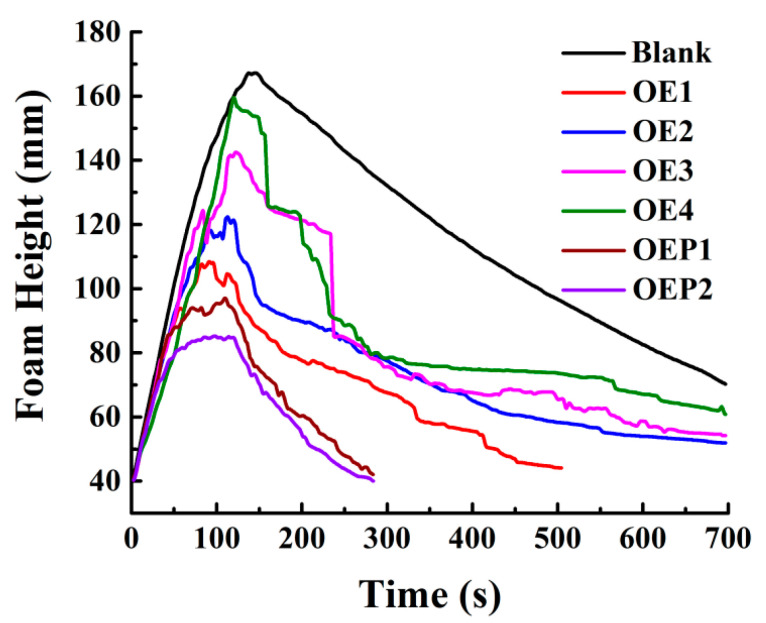
Changes in the foam heights of the samples of polycarboxylate superplasticizer (20 wt%) mixed with various defoamers (0.1 wt%) against time.

**Figure 6 materials-15-07492-f006:**
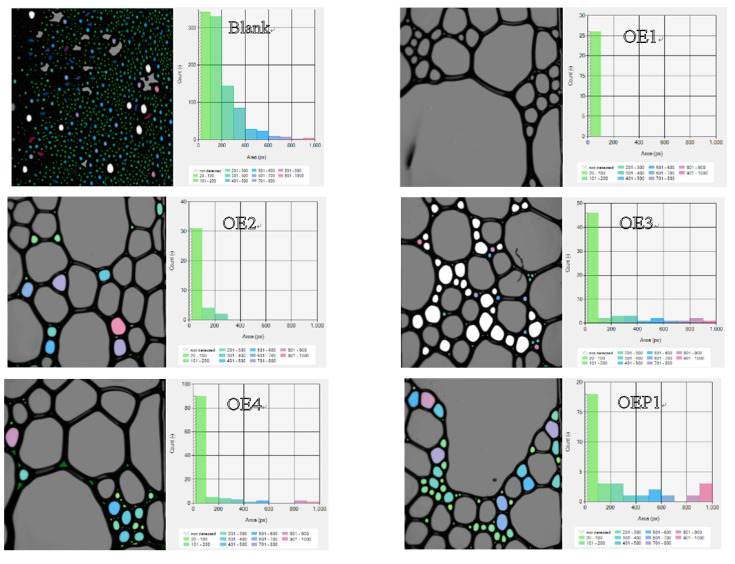
Photographs and size distribution columns of the solutions of the polycarboxylate superplasticizer mixed with various defoamers at the total time of 700 s.

**Figure 7 materials-15-07492-f007:**
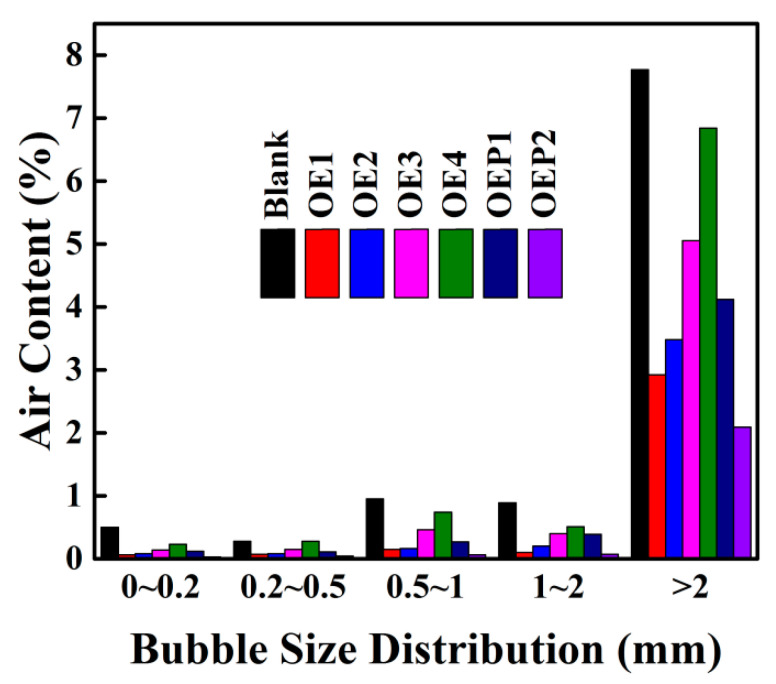
The bubble size distribution of the fresh mortar samples mixed with various defoamers.

**Figure 8 materials-15-07492-f008:**
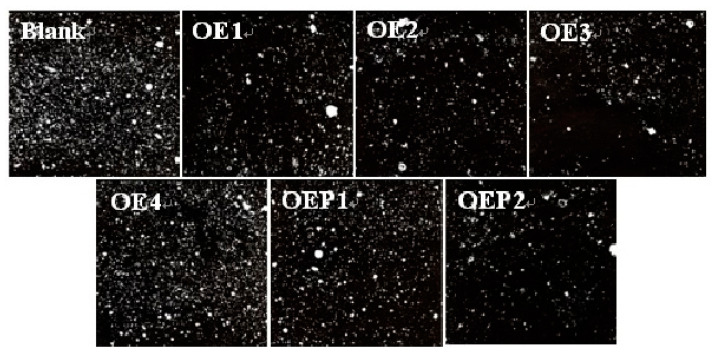
Air-void photographs of the hardened specimen samples mixed with various defoamers.

**Figure 9 materials-15-07492-f009:**
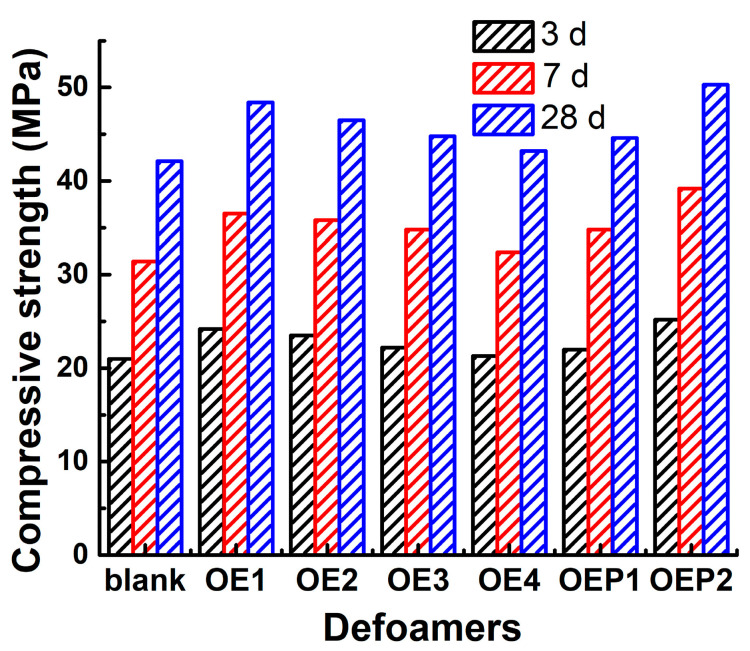
Compressive strengths of the hardened mortar samples after the incubation times of 3 (black), 7 (red) and 28 (blue) days mixed with various polyether-type defoamers.

**Table 1 materials-15-07492-t001:** Structure list of the prepared defoamers.

Defoamers	Chemical Structure
OE1	OA-4EO
OE2	OA-6EO
OE3	OA-8EO
OE4	OA-10EO
OEP1	OA-4EO-4PO
OEP2	OA-4PO-4EO

**Table 2 materials-15-07492-t002:** Surface activity parameters of the solutions mixed with various defoamers.

Defoamers	CMC (mM)	γCMC (mN/m)
OE1	0.007	30.63
OE2	0.011	31.53
OE3	0.016	35.12
OE4	0.025	35.89
OEP1	0.013	34.47
OEP2	0.004	27.30

**Table 3 materials-15-07492-t003:** Maximum foam heights and complete defoaming times of the samples of the polycarboxylate superplasticizer mixed with various defoamers.

Defoamers	Maximum Foam Height (mm)	Complete Defoaming Time (s)
Blank	167	>700
OE1	108	466
OE2	122	>700
OE3	143	>700
OE4	159	>700
OEP1	96	214
OEP2	85	164

**Table 4 materials-15-07492-t004:** The air content and spread diameters of the fresh mortar samples mixed with different defoamers.

Defoamers	Spread Diameter (mm)	Bulk Density (g/L)	Air Content (%)
Blank	202	1807	10.4
OE1	247	1985	3.3
OE2	237	1966	4.0
OE3	216	1912	6.2
OE4	211	1855	8.6
OEP1	228	1941	5.0
OEP2	252	2008	2.3

**Table 5 materials-15-07492-t005:** Performance of the hardened mortar samples mixed with various defoamers.

Defoamers	Air Content (%)	Air-Void Spacing Factors (mm)	Compressive Strength (MPa)		
			3 d	7 d	28 d
Blank	12.33	0.22	21	31.4	42.1
OE1	4.98	0.49	24.2	36.5	48.4
OE2	6.38	0.41	23.5	35.8	46.5
OE3	7.36	0.36	22.2	34.8	44.8
OE4	9.94	0.25	21.3	32.4	43.2
OEP1	7.73	0.33	22	34.8	44.6
OEP2	3.05	0.58	25.2	39.2	50.3

## Data Availability

The data are contained within the article. Additional data are available upon request from the corresponding author.

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
