# Peer review of "Preparation and Properties of Different Polyether-Type Defoamers for Concrete"

_materials, 2022, doi:10.3390/ma15217492_

Round 1

Reviewer 1 Report (Previous Reviewer 1)

The manuscript reported the preparation of polyether using oleyl alcohol (OA) as the starting agent and linked with ethylene oxide (EO) and propylene oxide (PO). The various defoaming properties in cement were studied such as surface tension, air contents, hardness, etc. The manuscript was written well.  Some minor comments should be addressed before publication.

1- It is highly recommended to redesign the abstract, especially the first sentence.

2- it is better to add more characteristic techniques to prove the material's performance.

3- it is better to check the manuscript language carefully. 

Author Response

1. Reviewer comments: It is highly recommended to redesign the abstract, especially the first sentence.

Our response: We appreciate the helpful comments from the reviewer. We have redesigned the Abstract Section following the reviewer’s advice.

2. Reviewer comments: It is better to add more characteristic techniques to prove the material's performance.

Our response: We appreciate the helpful comments from the reviewer. The chemical structure of the prepared defoamers was characterized by FT-IR, 1H-NMR and elemental analysis, the results clearly indicate that the defoamers of correct chemical structure and high purity were obtained. The properties of different defoamers were tested in aqueous solutions (surface activity and foaming property), fresh (air content and bubble size distribution) and hardened (air content, air-void parameter and compressive strength) cement mortars, the structure-property relationship of the defoamers was thus fully investigated.

3. Reviewer comments: It is better to check the manuscript language carefully.

Our response: We appreciate the helpful comments from the reviewer. We have checked the manuscript language carefully following the reviewer’s advice.

Reviewer 2 Report (New Reviewer)

The authors report on a series of polyether type defoamers for concrete which consist of the same hydrophobic part (alkyl chain) and different hydrophilic part (polyether chains), and study their defoaming capacity. The authors clam that the best results were achieved for the defoamer with ethylene oxide (EO) and propylene oxide (PO) units. The article provides new information and can be interested to the scientific community dealing with concrete. From my opinion, the article can be published in Materials after minor revisions.

Line 26: ref 38 is cited just after ref. 7.

Line 75: it should be “sodium hydride was dissolved” instead of “were dissolved”

Line 48: here and in many other places of the manuscript, hyphen should be used for ranges instead of tilde ~.

Line 155: it should be “samples” instead of “sample”

Line 193: “the results indicates” please, correct.

Line 221: it should be “bubble size” instead of “bubbles size”.

It is clear that the defoaming capacity becomes lower with higher number of EO units. Also, the defoaming capacity for the compounds with EO and PO units is better. It is explained by hydrophobicity and number of EO units. The authors claim that when “PO before EO” is used (OEP 2), it leads to the best results. However, it remains unclear why OEP 1 and OEP 2 show different results. Both of them (OEP 1 and 2) possess the same number of EO and PO units. Please, suggest an explanation why the results for OEP 1 and OEP 2 are different and add this explanation in the Results and Discussion and in the Conclusions.

Author Response

1. Reviewer comments: Line 26: ref 38 is cited just after ref. 7.

Our response: We appreciate the helpful comments from the reviewer. We have adjusted the reference numbers following the reviewer’s advice (Refs. 8 and 9 of the revised manuscript).

2. Reviewer comments: Line 75: it should be “sodium hydride was dissolved” instead of “were dissolved”.

Line 48: here and in many other places of the manuscript, hyphen should be used for ranges instead of tilde ~.

Line 155: it should be “samples” instead of “sample”

Line 193: “the results indicates” please, correct.

Line 221: it should be “bubble size” instead of “bubbles size”.

Our response: We appreciate the helpful comments from the reviewer. We have corrected these mistakes following the reviewer’s advice.

3. Reviewer comments: It is clear that the defoaming capacity becomes lower with higher number of EO units. Also, the defoaming capacity for the compounds with EO and PO units is better. It is explained by hydrophobicity and number of EO units. The authors claim that when “PO before EO” is used (OEP 2), it leads to the best results. However, it remains unclear why OEP 1 and OEP 2 show different results. Both of them (OEP 1 and 2) possess the same number of EO and PO units. Please, suggest an explanation why the results for OEP 1 and OEP 2 are different and add this explanation in the Results and Discussion and in the Conclusions.

Our response: We appreciate the helpful comments from the reviewer. From the chemical structure, polyether type defoamer is a class of amphiphilic surfactant containing both hydrophobic and hydrophilic parts. The hydrophobic part is usually a long alkyl chain, and the hydrophilic part is a polyether chain consisting of repeat EO or PO units (Refs. 18 and 19). Since PO is more hydrophobic than EO, which exhibits weak hydrophilicity. Compared with OEP1 (OA-4EO-4PO, hydrophobic-hydrophilic-weakly hydrophilic), the structure of OEP2 (OA-4PO-4EO, hydrophobic-weakly hydrophilic-hydrophilic) is closer to that of an amphiphilic surfactant, thus OEP2 exhibits stronger defoaming capacity. We have added this explanation in the 3.3 of Results and Discussion Section and in the Conclusions Section of the revised manuscript.

Reviewer 3 Report (New Reviewer)

The paper has not well written. A major revisions is needed.

-some main parts of the paper must be rewritten; Abstract, Conclusions. Also, some sections should be improved; conclusions, discussions and results.

-More new and relevant papers should be added to the paper.

-A section titled "Methodology" with high quality should be added to the paper. In this section, the method of study should be explained.

- What is the novelty of the paper? It should be highlighted at the end of introduction.

-The results should be classified and described with a specific purpose. The process in the article is very confusing and ambiguous.

Author Response

1. Reviewer comments: Some main parts of the paper must be rewritten; Abstract, Conclusions. Also, some sections should be improved; conclusions, discussions and results.

Our response: We appreciate the helpful comments from the reviewer. We have carefully rewritten and improved the paper following the reviewer’s advice.

2. Reviewer comments: More new and relevant papers should be added to the paper.

Our response: We appreciate the helpful comments from the reviewer. We have added more new and relevant papers as the references (Refs. 8, 9, 40, 41 and 42 of the revised manuscript).

3. Reviewer comments: A section titled "Methodology" with high quality should be added to the paper. In this section, the method of study should be explained.

Our response: We appreciate the helpful comments from the reviewer. We have added the section titled "Methodology" to the paper (2.2 of the Materials and Methods Section of the revised manuscript).

4. Reviewer comments: What is the novelty of the paper? It should be highlighted at the end of introduction.

Our response: We appreciate the helpful comments from the reviewer. We have highlighted the novelty of the study more clearly at the end of Introduction Section, and the differences of this study from the published study in literature have also been introduced (the Introduction Section in the revised manuscript).

5. Reviewer comments: The results should be classified and described with a specific purpose. The process in the article is very confusing and ambiguous.

Our response: We appreciate the helpful comments from the reviewer. We have carefully rewritten and improved the Results and Discussion Section following the reviewer’s advice.

Round 2

Reviewer 3 Report (New Reviewer)

The paper has been revised according to the reviewers comments.

This manuscript is a resubmission of an earlier submission. The following is a list of the peer review reports and author responses from that submission.

Round 1

Reviewer 1 Report

The manuscript reported the preparation of polyether using oleyl alcohol (OA) as the starting agent and linked with ethylene oxide (EO) and propylene oxide (PO). The various defoaming properties in cement were studied such as surface tension, air contents, hardness, and so on. The work novelty has not been expressed in the work.

Please see the following comments

1-    In the abstract section, the authors have not expressed the work's novelty. It is better to address the reason for a defoaming activity for polyether linked with EO and PO units.

2-    The end of the introduction is repeated data as abstract. It is better to redesign this part.

3-    In the various defoamer preparation, the authors prepared OE and OEP, what about the possibility of synthesizing OP (OA+ PO)? It is better to address the different polyether structures.

4-    The H-NMR interpretation is not sufficient to prove the structural change and formation of new defoamers. It is highly recommended to redesign this section.

5-    It is highly recommended to add more characterization techniques to support the prepared polyether such as SEM images, melting point, Raman shift, and contact angle.

6-    In Figure 3, the surface tension of OE1 is lower than OE, even though the EO value of both defoamers is the same. can the authors explain why?

Author Response

1. Reviewer comments: In the abstract section, the authors have not expressed the work's novelty. It is better to address the reason for a defoaming activity for polyether linked with EO and PO units.

Our response: We appreciate the helpful comments from the reviewer. We have expressed the novelty of this work (the Abstract and Introduction Sections in the revised manuscript).

From the chemical structure, polyether type defoamer is a class of amphiphilic surfactant containing both hydrophilic and hydrophobic parts. The hydrophilic part is a polyether chain consisting of repeat EO or PO units, and the hydrophobic part is usually a long alkyl chain. In our previous study, we reported the polyether type defoamers consisting of both hydrophobic alkyl chains and hydrophilic EO or PO units, which exhibited excellent defoaming performance [16,17]. We have addressed the reason for a defoaming activity for polyether linked with EO and PO units (the Introduction Section in the revised manuscript).

2. Reviewer comments: The end of the introduction is repeated data as abstract. It is better to redesign this part.

Our response: We appreciate the helpful comments from the reviewer. We have redesigned the end of the Introduction Section in the revised manuscript.

3. Reviewer comments: In the various defoamer preparation, the authors prepared OE and OEP, what about the possibility of synthesizing OP (OA+PO)? It is better to address the different polyether structures.

Our response: We appreciate the helpful comments from the reviewer. The chemical structure of OEP2 in this work is OA-4PO-4EO (Table 1) which contains the structure of OA-PO, thus synthesizing OP (OA+PO) is totally possible. We have addressed the chemical structure of different polyethers (Figure 1 of the revised manuscript).

4. Reviewer comments: The H-NMR interpretation is not sufficient to prove the structural change and formation of new defoamers. It is highly recommended to redesign this section.

Our response: We appreciate the helpful comments from the reviewer. We have added the elemental analysis results of the defoamers (2.2 of the Materials and Methods Section in the revised manuscript). The chemical structure of the defoamers was characterized by FT-IR, 1H-NMR and elemental analysis, the results indicate that the defoamers of correct chemical structure and high purity were obtained. We have redesigned the section of 1H-NMR interpretation (3.1 of the Results and Discussion Section in the revised manuscript).

5. Reviewer comments: It is highly recommended to add more characterization techniques to support the prepared polyether such as SEM images, melting point, Raman shift, and contact angle.

Our response: We appreciate the helpful comments from the reviewer. We have added the elemental analysis characterization to support the prepared polyether (2.2 of the Materials and Methods Section in the revised manuscript). The defoamers were characterized by FT-IR, 1H-NMR and elemental analysis, the results indicate that the defoamers of correct chemical structure and high purity were obtained. In addition, all the defoamers are liquids under room temperature, their melting points are all below 4 oC.

6. Reviewer comments: In Figure 3, the surface tension of OE1 is lower than OEP1, even though the EO value of both defoamers is the same. Can the authors explain why?

Our response: We appreciate the helpful comments from the reviewer. The air content results indicated that the defoamer containing less EO amount (higher hydrophobicity) had higher defoaming capacity, and the defoamer linked with PO before EO (OEP2) had strongest defoaming capacity (Table 4). The surface tension results indicated that the surface tension decreased with the decrease of EO amount (Figure 4 of the revised manuscript). The results of air content are consistent with the results of surface tension, the defoamer having higher defoaming capacity (OEP2>OE1>OE2>OEP1>OE3>OE4) showed lower surface tension value. OE1 showed higher defoaming capacity compared with OEP1, thus had lower surface tension.

Reviewer 2 Report

Comments

This paper studied Properties of Different Polyether Type Defoamers for Concrete. The outcome of the paper is interesting however, there are several aspects that need to be improved. The reviewer can only recommend for publication if the author satisfactorily address the following major comments in the revised version.

1.       The strength values in Table 5 could have been presented in graphs.

2.       The research questions and justification of selecting variable parameters should be highlighted.

3.       Which test standards was considered in this study? How many replicate samples were tested in each category?

4.       The failure mechanism of the specimen should be discussed more clearly.

5.       The novelty of the study should be highlighted more clearly at the end of introduction section. How this study is different from the published study in literature?

6.       How the outcome of this study will benefit researchers and end users? This need to be highlighted in introduction or end of conclusion.

7.       The properties of concrete is interesting but not fully novel. Therefore, the recent study in this area should be discussed in introduction section to improve the background information. Recently, fresh concrete properties was studied in [Ref: Characteristics, strength development and microstructure of cement mortar containing oil-contaminated sand] while the hardened concrete properties was studied in [Ref: Recycling of landfill wastes (tyres, plastics and glass) in construction–A review on global waste generation, performance, application and future opportunities]. Suggest to include them in introduction section with proper citations to improve the background study.

I would be happy to see the revised version to understand how these comments are being addressed.

Author Response

1. Reviewer comments: The strength values in Table 5 could have been presented in graphs.

Our response: We appreciate the helpful comments from the reviewer. We have presented the strength values of Table 5 in graphs (Figure 9 of the revised manuscript).

2. Reviewer comments: The research questions and justification of selecting variable parameters should be highlighted.

Our response: We appreciate the helpful comments from the reviewer. We have highlighted the research questions and justification of selecting variable parameters (end of the Introduction Section in the revised manuscript).

3. Reviewer comments: Which test standards was considered in this study? How many replicate samples were tested in each category?

Our response: We appreciate the helpful comments from the reviewer. Surface activity, foam analysis, bubble size distribution and air-void parameter were tested as the previous report (Refs. 16 and 17). Spread diameters were tested according to the Chinese National Standard GB/T 8077-2012 13. Air contents were tested according to the Chinese National Standard GB/T 50080- 2016 15. Compressive strengths were tested according to the Chinese National Standard GB/T 50081-2002 6.

For each category, three same samples were prepared and tested, and the average value was recorded. All the tests showed good replicates. We have added these sentences following the reviewer’s advice (2.4 and 2.5 of the Materials and Methods Section in the revised manuscript).

4. Reviewer comments: The failure mechanism of the specimen should be discussed more clearly.

Our response: We appreciate the helpful comments from the reviewer. We have discussed the failure mechanism of the specimens more clearly following the reviewer’s advice (3.5 of the Results and Discussion Section in the revised manuscript).

5. Reviewer comments: The novelty of the study should be highlighted more clearly at the end of introduction section. How this study is different from the published study in literature?

Our response: We appreciate the helpful comments from the reviewer. We have highlighted the novelty of the study more clearly at the end of introduction section, the differences of this study from the published study in literature have been introduced (the Introduction Section in the revised manuscript).

6. Reviewer comments:How the outcome of this study will benefit researchers and end users? This need to be highlighted in introduction or end of conclusion.

Our response: We appreciate the helpful comments from the reviewer. We have highlighted the outcome and benefit of this study in introduction or end of conclusion (the Introduction and Conclusion Section in the revised manuscript).

7. Reviewer comments: The properties of concrete is interesting but not fully novel. Therefore, the recent study in this area should be discussed in introduction section to improve the background information. Recently, fresh concrete properties was studied in [Ref: Characteristics, strength development and microstructure of cement mortar containing oil-contaminated sand] while the hardened concrete properties was studied in [Ref: Recycling of landfill wastes (tyres, plastics and glass) in construction–A review on global waste generation, performance, application and future opportunities]. Suggest to include them in introduction section with proper citations to improve the background study.

Our response: We appreciate the helpful comments from the reviewer. We have cited these references in the introduction section following the reviewer’s advice (Refs. 38 and 39 in the revised manuscript).